# NEURAL PARTIAL DIFFERENTIAL EQUATIONS WITH FUNCTIONAL CONVOLUTION

## ABSTRACT

We present a lightweighted neural PDE representation to discover the hidden structure and predict the solution of different nonlinear PDEs. Our key idea is to leverage the prior of "translational similarity" of numerical PDE differential operators to drastically reduce the scale of learning model and training data. We implemented three central network components, including a neural functional convolution operator, a Picard forward iterative procedure, and an adjoint backward gradient calculator. Our novel paradigm fully leverages the multifaceted priors that stem from the sparse and smooth nature of the physical PDE solution manifold and the various mature numerical techniques such as adjoint solver, linearization, and iterative procedure to accelerate the computation. We demonstrate the efficacy of our method by robustly discovering the model and accurately predicting the solutions of various types of PDEs with small-scale networks and training sets. We highlight that all the PDE examples we showed were trained with up to 8 data samples and within 325 network parameters.

## 1 INTRODUCTION

**Problem definition**    We aim to devise a learning paradigm to solve the inverse PDE identification problem. By observing a small data set in the PDE's solution space with an *unknown* form of equations, we want to generate an effective neural representation that can precisely reconstruct the hidden structure of the target PDE system. This neural representation will further facilitate the prediction of the PDE solution with different boundary conditions. The right inset figure shows a typical example of our target problem: by observing a small part (4 samples in the figure) of the solution space of a nonlinear PDE system $\mathcal{F}(\boldsymbol{x}) = b$, *without knowing its analytical equations*, our neural representation will depict the hidden differential operators underpinning $\mathcal{F}$ (e.g., to represent the unknown differential operator $\nabla \cdot (1 + x^2)\nabla$ by training the model on the solution of $\nabla \cdot (1 + x^2)\nabla x = b$.

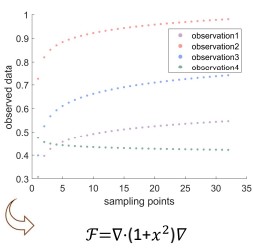

$$\mathcal{F} = \nabla \cdot (1 + x^2)\nabla$$

**Challenges to solve**    The *nonlinearity* and the *curse of dimensionality* of the target PDE's solution manifold are the two main challenges for the design of a high-performance neural discretization. An effective neural representation of a PDE system plays an essential role to solve these challenges. In retrospect, the design of neural PDE representations has been evolving from the raw, unstructured networks (e.g., by direct end-to-end data fitting) to various structured ones with proper mathematical priors embedded. Examples include the residual-based loss function (e.g., physics-informed networks Raissi et al., 2020; Lu et al., 2019; Raissi et al., 2019), learnable convolution kernels (e.g., PDE-Nets Long et al., 2018a;b; 2019), and hybrid of numerical stencils and MLP layers (e.g., see Amos & Kolter, 2017; Pakravan et al., 2020; Geng et al., 2020; Stevens & Colonius, 2020). Following this line of research, we aim to devise a lightweighted neural PDE representation that fuses the mathematical equation's essential structure, the numerical solvers' computational efficiency, and the neural networks' expressive power. In particular, we want to aggressively reduce the *scale* of both model parameters and training data to some extremal extent, while extending the *scope* of the targeted PDE systems to a broad range, encompassing equations that are both linear and nonlinear, both steady-state and dynamic.

**Translational similarity of differential operators** Our neural PDE representation design is motivated by the historical successes of the various *sparse*, *iterative* numerical solvers in solving non-linear PDEs over the past decades. The key observation we have made is that the efficacy of a classical numerical PDE solver relies on the *translational similarity* of its discretized, local differential operators. Namely, the form of a differential operator can be written as a functional $C(x, p)$ with respect to the the PDE unknown $x$ and

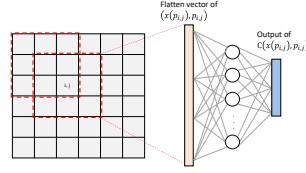

the *local* position $p$, which is showed in the right inset figure. For example, for a linear Poisson system $\nabla \cdot \nabla x = b$, $C$ is a constant function; for a linear Poisson system with embedded boundary conditions, $C$ is a function of position $p$; for a nonlinear PDE $\nabla \cdot (1 + x^2)\nabla x = b$, $C$ is a function of PDE unknown $x$ (or both $x$ and $p$ if it has embedded boundaries). For most numerical PDEs, these local functional operators can be parameterized and built *on-the-fly* within the solver iterations. Such operators' locality further inspired the design of a variety of computationally efficient PDE solvers, among which the most famous one is the matrix-free scheme that has been used widely in solving large-scale physical systems on GPU. These local procedures for stencil creation have demonstrated their extreme performance in accommodating PDE solvers. From a machine learning perspective, these "translational similar" differential operators resemble the concept of convolution operators that function as the cornerstone to embed the "translational invariant" priors into neural networks ( see LeCun et al., 1995; 1998).

**Method overview** In this work, **we leverage the PDE differential operators' "translational similarity" in a reverse manner by devising a local neural representation that can uncover and describe the global structure of the target PDE**. At the heart of our approach lies in a differential procedure to simultaneously describe the spatial coupling and the temporal evolution of a local data point. Such procedure is implemented as a parameterized micro network, which is embedded in our iterative solving architecture, to learn the numerical process of converging from an initial guess to a final steady state for a PDE solution. We name these embedded micro networks "*functional convolutions*," for two reasons. First, fitting the parameters of these local embedded networks amounts to the exploration of the optimal function that best describes the observed solution of the unknown nonlinear PDE within a functional space. Second, the local differential operators that span this functional space can be treated as numerically employing convolution kernels Hsieh et al. (2018); Lin et al. (2013). Based on these functional convolutions, we are able to devise a learning framework by embedding the micro network architecture within an iterative procedure to 1) backwardly learn the underpinning, spatially varying structures of a nonlinear PDE system by iteratively applying the adjoint linear solvers and 2) forwardly predict the steady states of a PDE system by partially observing data samples of its equilibrium. We show that our model can simultaneously discover structures and predict solutions for different types of nonlinear PDEs. We particularly focus on solving elliptic boundary value problems that were less explored in the current literature.

## 2 MOTIVATING EXAMPLE: FORWARD NUMERICAL PDE

**Naming convention** We first show a motivating example to demonstrate the standard process of a *forward* numerical PDE solver. We take the simplest Poisson equation with Dirichlet boundary conditions as an example. The mathematical equation of a Poisson system can be written as $\nabla \cdot \nabla x = b$ for $x \in \Omega$, with $x$ as the PDE unknowns, $b$ as the right-hand side, and $\Omega$ as the problem's domain. The boundary conditions are enforced in a Dirichlet way (by assigning values directly) as $x = \hat{x}$ on the domain boundary, with $\hat{x}$ as the specified boundary values. To create a discretized, numerical system to solve the equation, we use the symbol $p$ to denote the **position** within the domain. **The numerical solution of the PDE amounts to seeking an unknown function $x(p)$ that can specify the value of $x$ in an arbitrary position $p$ within $\Omega$.**

**Linear PDE** As shown in Figure 1, we illustrate how to solve the Poisson system using a finite-difference method. We first subdivide the domain into $n$ cell (segment intervals in 1D and squares in 2D) with the cell size of $\Delta p$. Taking the 2D case for example, we can derive the discretized Poisson equation by approximating the Laplacian operator on each grid cell using the central finite difference method $(-x_{i-1,j} - x_{i+1,j} + 4x_{i,j} - x_{i,j-1} - x_{i,j+1})/\Delta p^2 = b_{i,j}$. The discretization of each cell forms one row in the linear system, and the combination of all the rows (cells) forms a sparse

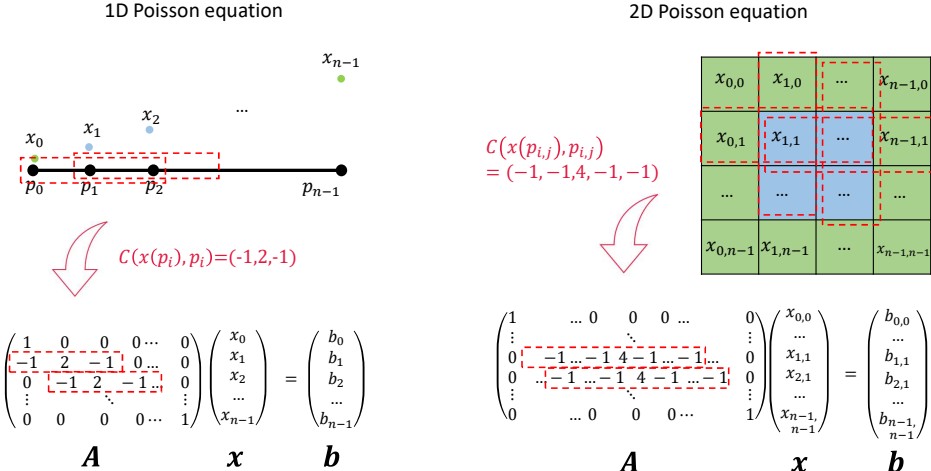

Figure 1: Schematic illustration to numerically solve the Poisson equation.

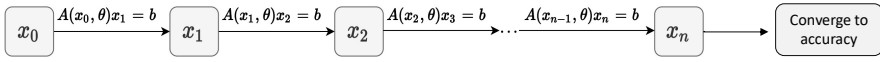

Figure 2: Forward data flow in a neural Picard solver to predict the solutions of a non-linear PDE.

linear system $A\boldsymbol{x} = \boldsymbol{b}$ to solve. For a linear Poisson system, each row of $A$ can be translated into a convolutional stencil instantiated with constant parameters, e.g., (-1,2,1) in 1D and (-1,-1,4,-1,-1) in 2D. This convolution perspective can be used to accelerate the numerical solver by maintaining a "conceptual" $A$ matrix without storing the redundant matrix element values in memory (we refer the readers to the matrix-free method Carey & Jiang (1986) for more details). This matrix-free nature indicates an extremely concise representation of a numerical Poisson system (i.e., the matrix A)—using a $1 \times 3$ or $3 \times 3$ convolution kernel with fixed values. **We use the symbol $C$ to denote the convolutional representation of** $A$**.** For a linear system, $C$ is independent from the values of $\boldsymbol{p}$ and $\boldsymbol{x}$.

**Nonlinear PDE and Picard interation** The nonlinear case of the Poisson system is a bit complicated. We can still use the matrix-free representation to describe a nonlinear Poisson system, but the parameters of this convolutional stencil now depends on both the local position $\boldsymbol{p}$ and the local unknown $\boldsymbol{p}(\boldsymbol{x})$. This dependency is nonlinear and therefore we cannot find the solution by solving a single $A\boldsymbol{x} = \boldsymbol{b}$. Here we present an iterative scheme—the Picard method—to solve a numerical nonlinear PDE system. (see Picard, 1893; Bai, 2010) Let's consider the nonlinear Poisson equation as: $\nabla \cdot \alpha(\boldsymbol{x}) \nabla \boldsymbol{x} = \mathbf{b}$ for $\boldsymbol{x} \in \Omega$ and Dirichlet boundary conditions on the boundary. The source of nonlinearity in this PDE includes the coefficient $\alpha(\boldsymbol{x})$ which is dependent on the solution of $\boldsymbol{x}$, e.g., $\alpha(\boldsymbol{x}) = 1 + \boldsymbol{x}^2$. A simple and effective fixed-point procedure to discretize and solve the nonlinear Poisson equation, named Picard iteration, can be sketched as follows:

$$\text{while: } |\boldsymbol{x}_n - \boldsymbol{x}_{n-1}| > \epsilon$$
$$\boldsymbol{x}_{n+1} = A^{-1}(\boldsymbol{x}_n)\boldsymbol{b}, \tag{1}$$

with the matrix $\boldsymbol{A}$ representing the current discretized nonlinear Laplacian operator approximated by the value of unknowns from the previous iteration. The key idea is to employ a linear approximation of a nonlinear problem and enforce such approximation iteratively to evolve to a steady state (see Figure 2). To uncover the underlying structure of $\boldsymbol{A}$, which can evolve both spatially and temporally, we make a prior assumption that $\boldsymbol{A}$ **can be described by a kernel function** $C(\boldsymbol{x}(\boldsymbol{p}), \boldsymbol{p})$. Such prior applies to most of the elliptic PDE systems where the spatial terms can be expressed as the combination of one or several differential operators. From a numerical perspective, $C$ describes the local discretized interaction between an element and its neighbours. It amounts to a function that returns all the non-zero elements for each row $i$ in $\boldsymbol{A}$ (think of $\boldsymbol{A}$ in a matrix-free way).

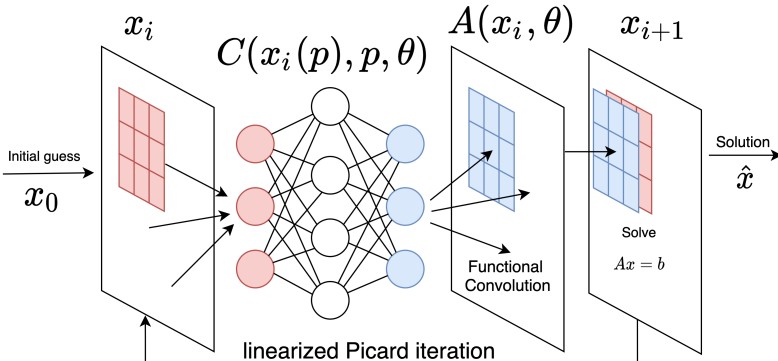

Figure 3: Overview: To obtain the solution of $A(x,\theta)x = b$, we first generate a random initial guess $x_0$. Then in each iteration step $i$, we apply the functional convolution $C$ on $x_i$ to obtain the convolution kernels $A(x_i, \theta)$. We obtain $x_{i+1}$ by solving $A(x_i, \theta)x = b$. We repeat this process to obtain a converged solution.

## 3 METHOD: BACKWARD NEURAL PDE

**Overview**  In this section, we present our neural PDE design motivated by the forward Picard solver with convolutional representation of the nonlinear system. Our framework consists of three key components: the neural representation of the convolution operator, the embedded Picard forward iterative solver, and the adjoint backward derivatives. The key idea is to differentiate the forward nonlinear Picard solver and build the neural representation for its sparse linearization step. This differentiation is implemented by our functional convolution scheme on the linearization level and the adjoint Picard for the nonlinear iterations.

### 3.1 FUNCTIONAL CONVOLUTION

In a conventional ML method, $C$ can be approximated by the combination of a set of kernels Long et al. (2018a) or by solving a deconvolution problem Xu et al. (2014); Jin et al. (2017); Zhang et al. (2017). However, these strategies do not suit our scenario, where the instant linear effects of the system should be approximated by extracting the nonlinear effects of $C$. A natural choice to approximate this spatially-and-temporally varying kernel function is to devise a neural network, which takes the form of $C(\boldsymbol{x}(\boldsymbol{p}), \boldsymbol{p}, \theta)$, with $\theta$ as the network parameters. Numerically, the global matrix $\boldsymbol{A}$ can be fully parameterized by $C(\boldsymbol{x}(\boldsymbol{p}), \boldsymbol{p}, \theta)$ by assuming the fact that $C$ is a non-weight-shared convolution operator in the spatial domain. As illustrated in Figure 3, such neural network can be further incorporated into a conventional nonlinear Picard solver to obtain the forward steady-state solution by solving the linear system $\boldsymbol{A}(\boldsymbol{x}_n, \theta)\boldsymbol{x}_{n+1} = \boldsymbol{b}(\boldsymbol{x}_n)$, where a black-box sparse linear solver can be used as in a conventional simulation program. The formal definition of functional convolution written in the kernel way is

$$\boldsymbol{A}(\boldsymbol{x}(p_i), \theta) = \sum_{p_j \in \mathcal{N}(p_i)} [\boldsymbol{C}(\boldsymbol{x}(\mathcal{N}(p_i)), \mathcal{N}(p_i), \theta)]\boldsymbol{e}(p_j) \tag{2}$$

where $\boldsymbol{A}(\boldsymbol{x}(p_i), \theta)$ is the $i^{th}$ row of matrix $\boldsymbol{A}$, $\mathcal{N}(p_i)$ is the neighboring positions of $p_i$, $\boldsymbol{x}(\mathcal{N}(p_i))$ is all the neighboring elements (all channels) around the position $p_i$ and $\boldsymbol{e}(p_j)$ is the unit vector representing the $j^{th}$ column in $\boldsymbol{A}$.

To specify the 2D example, equation (2) has the following form

$$\boldsymbol{A}(\boldsymbol{x}_{m,n}, \theta) = \sum_{i=-1}^{1} \sum_{j=-1}^{1} [\boldsymbol{C}(\mathcal{N}(x_{m,n}), \theta)]_{i,j} \boldsymbol{e}_{m+i, n+j} \tag{3}$$

where $x_{m,n}$ is the element that lies in row $m$, column $n$ of the feature map. The input $\mathcal{N}(x_{m,n}) = \{x_{m+i,n+j}$ for $i, j = -1, 0, 1\}$ is the flatten vector of neighboring elements of pixel $x_{m,n}$. After a

simple neural network $C$ with parameters $\theta$, we obtain the output $C(\mathcal{N}(x_{m,n}), \theta)$, which is a vector with the same length of $\mathcal{N}(x_{m,n})$.

## 3.2 ADJOINT DERIVATIVES

To train our functional convolution operator in an end-to-end fashion, we calculate the derivatives of the loss function $L$ regarding the network parameters $\theta$ following the chain rule. For a neural PDE with $n$ layers for the outer Picard loop (see Figure 3), we can calculate the sensitivity $\partial L / \partial \theta$ as:

$$\frac{\partial L}{\partial \theta} = \frac{\partial L}{\partial \boldsymbol{x}_n}\frac{\partial \boldsymbol{x}_n}{\partial \theta} + \frac{\partial L}{\partial \boldsymbol{x}_n}\frac{\partial \boldsymbol{x}_n}{\partial \boldsymbol{x}_{n-1}}\frac{\partial \boldsymbol{x}_{n-1}}{\partial \theta} + \cdots + \frac{\partial L}{\partial \boldsymbol{x}_n}\frac{\partial \boldsymbol{x}_n}{\partial \boldsymbol{x}_{n-1}}\cdots\frac{\partial \boldsymbol{x}_2}{\partial \boldsymbol{x}_1}\frac{\partial \boldsymbol{x}_1}{\partial \theta} \tag{4}$$

For layer $i$ that maps from $\boldsymbol{x}_i \to \boldsymbol{x}_{i+1}$ by solving $\boldsymbol{A}(\boldsymbol{x}_i, \theta)\boldsymbol{x}_{i+1} = \boldsymbol{b}$, we can express its backward derivative $\boldsymbol{y}_i$ as

$$\boldsymbol{y}_i = \frac{\partial \boldsymbol{x}_{i+1}}{\partial \boldsymbol{x}_i} = -\boldsymbol{A}^{-1}(\boldsymbol{x}_i, \theta)\frac{\partial \boldsymbol{A}(\boldsymbol{x}_i, \theta)}{\partial x_i}\boldsymbol{x}_{i+1}, \tag{5}$$

which can be solved by two linear systems, including a forward equation and a backward adjoint equation:

$$\boldsymbol{A}(\boldsymbol{x}_i, \theta)\boldsymbol{x}_{i+1} = b \tag{6a}$$

$$\boldsymbol{A}(\boldsymbol{x}_i, \theta)\boldsymbol{y}_i = \frac{\partial \boldsymbol{A}(\boldsymbol{x}_i, \theta)}{\partial \boldsymbol{x}_i}\boldsymbol{x}_{i+1} \tag{6b}$$

Similarly, we can get $\partial L / \partial \theta$ as:

$$\boldsymbol{z}_i = \frac{\partial \boldsymbol{x}_{i+1}}{\partial \theta} = -\boldsymbol{A}^{-1}(\boldsymbol{x}_i, \theta)\frac{\partial \boldsymbol{A}(\boldsymbol{x}_i, \theta)}{\partial x_i}\boldsymbol{x}_{i+1} \tag{7}$$

which can be calculated by solving one additional adjoint linear system:

$$\boldsymbol{A}(\boldsymbol{x}_i, \theta)\boldsymbol{z}_i = \frac{\partial \boldsymbol{A}(\boldsymbol{x}_i, \theta)}{\partial \theta}x_{i+1} \tag{8}$$

---

**Algorithm 1** Backward derivative of a nonlinear PDE boundary value problem

---

**Input:** $\boldsymbol{x}_1, \boldsymbol{b}, C, L$
**Output:** $\partial L / \partial \theta$
  //Forward linearization:
  **for** $i = 0 \to N - 1$ **do**
    Solve $\boldsymbol{A}(\boldsymbol{x}_i, \theta)x_{i+1} = \boldsymbol{b}$;
  **end for**
  //Backward adjoint:
  **for** $i = N - 2 \to 1$ **do**
    Solve adjoints (6b) and (7) for $\partial \boldsymbol{x}_{i+1} / \partial \theta$
    $\partial L / \partial \theta \mathrel{+}= (\partial L / \partial \boldsymbol{x}_{i+1})(\partial \boldsymbol{x}_{i+1} / \partial \theta)$
  **end for**

---

To calculate the terms $\partial \boldsymbol{A}/\partial \boldsymbol{x}_i$ and $\partial \boldsymbol{A}/\partial \theta$ in the chain, we take advantage of the sparse nature of $\boldsymbol{A}$ by calculating $\partial C/\partial \boldsymbol{x}_i$ and $\partial C/\partial \theta$ first and then distribute their values into the non-zero entries of the global matrix. Because $C(\boldsymbol{x}(\boldsymbol{p}), \boldsymbol{p}, \theta)$ is a functional convolution operator represented by a standard neural network, its derivatives regarding the first-layer input $\boldsymbol{x}$ and the parameters $\theta$ can be calculated in a straightforward way by the auto-differentiation mechanism. The overall algorithm is summarized as in Algorithm 1.

## 4 NETWORK ARCHITECTURE AND DATA TRAINING

The network we use in this study is summarized in Table 2. We use a hybrid method of IpOpt Wächter (2009) and Adam optimizer to optimize the parameters int the neural networks. Despite its fast converging rate, IpOpt optimizer typically converges to a local minimum or fail to converge due to a bad initialization. By introducing the hybrid optimization method, we are able to solve the problem. The optimized parameters from IpOpt are then oscillated and refined by Adam optimizer for a certain number of iterations. In case that the loss not converge, we then use the optimized parameters from Adam as the initial guess to warm-start the IpOpt and repeat this IpOpt-Adam process until the loss converges to the tolerance. For the Adam optimizer, we set the parameters to

be $learning\ rate = 1e - 3$, $\beta_1 = 0.9$, $\beta_2 = 0.999$, $\epsilon = 1e - 8$. We compare the performance of our hybrid optimization method with Adam and a typical SGDM method in solving the 1D Poisson problem in Figure 6. The results show that the hybrid method not only obtains a faster converging rate, but also converges to an extremely small loss compared with other methods.

## 5 RESULTS

We test the efficacy of our method by using it to fit a set of PDEs, ranging from the standard linear Poisson's problem to highly nonlinear temporally evolving PDEs. We build our neural network architecture by implementing both the differentiable Picard iterations and a simple fully-connected network in C++ 17. The training process for all the examples was run on an i7-8665 four-core desktop. The PDE and boundary conditions, sample size, and test results were summarized in Table 2 in Appendix.

### 5.1 1D EXAMPLES

**Constant kernels**    We first test our framework by fitting the underlying structure of a standard 1D Poisson problem $\nabla \cdot \nabla \boldsymbol{x} = b$ with Dirichlet boundary conditions. Here the target true solution is set to be $x = (ap)^2$, with $p$ as the spatial coordinate of the data sample and $a$ as a coefficient that varies for different data sample. The training dataset consists of four sample points, observed from the two solutions of the equations parameterized with different values of $a$ and boundary conditions. The 4-size training data is sampled on a $128 \times 1$ grid with two layers of picard network. After training, we run 16 PDE tests with varying $a$ and boundary conditions and obtain an averaged MSE loss as 8.3e-20. The predicted solutions in all 16 test cases match the analytical curves with the maximum MSE 1.5e-14, as shown in in Figure 9a.

Next, we test the same framework on another two Poisson problems. The first problem is $\nabla \cdot \nabla \boldsymbol{x} = 0$ with Neumann boundary conditions which does not have an analytical solution (see Figure 9b). The prediction results from our model are shown in Figure 9b with an averaged MSE 5.0e-29. The other problem is with the target solution as $x = sin(ap)$. The results are shown in Figure 4 with the 9.8e-4 MSE loss compared with the true solution.

**Stability test with noisy input**    We further conduct a stability test by adding noise values to the setting of the 1D constant kernel experiments. The training dataset consists of two sample points, observed from the two noisy solutions of the equations. We test our framework with various scales of noise from $[-.1, .1]$ to $[-.35, .35]$ with step size of .05 with respect to the extreme values in the target solution. The training data is sampled on a $32 \times 1$ grid with two layers of picard network. We compare our framework with denoised solutions in Figure 10. The figure shows the framework can automatically obtain precise solution even though our framework cannot access accurate solutions during the training procedure.

**Spatially varying coefficients**    Next we test our model by predicting the solutions of a Poisson PDE with spatially varying coefficients. The PDE has the analytical formula $\nabla \cdot (1 + |\pi p|)\nabla \boldsymbol{x} = 0$, with $p$ as the spatial coordinate of the data sample. We build the training dataset by solving the equation with randomly set boundary conditions on a $32 \times 1$ grid. The data sample size is 4 and each data sample has the full observation of the solution space. The input of the network is the current values of $\boldsymbol{x}$ and the sample positions $\boldsymbol{p}$. We show that our model can precisely uncover the distribution of the hidden latent space behind the Laplacian operator with spatially varying coefficients by fitting a functional convolution operator that predicts the solutions of the PDE in the forward process (see Figure 3). The average MSE loss is 1.3e-06 for this case.

**Non-linear equations**    In this example, we demonstrate the ability of our neural PDE solver by solving a non-linear PDE with the form: $\nabla \cdot (1 + |x| + sin(|x| * .001))\nabla x = 0$ Dirichlet boundary conditions are set on the two endpoints of the domain. The training dataset is generated by solving the PDE with standard Picard iterations. The number of the neural PDE network layers is set to be 5. We employ 4 solution samples on a $32 \times 1$ discretization for training. As shown in Figure 11, our model can precisely uncover the hidden nonlinear structures of the PDE kernel and predicts the numerical solution by employing the learned functional convolution through the Picard iterations.

## 5.2 2D EXAMPLES

**Poisson equation** We then expand our model to solve 2D problems. Similarly, we start from employing our model to predict the solutions for two standard 2D Poisson problems $\nabla \cdot \nabla \boldsymbol{x} = b$ with Dirichlet boundary conditions, whose target true solutions are set to be $\boldsymbol{x} = (ap_u)^3 + ap_v^2$ and $\boldsymbol{x} = sin(ap_u + ap_v)$ respectively. Here $p_u$ and $p_v$ refer to the $x$ axis and $y$ axis coordinates of the data sample. We use 4-size data samples, which are sampled on a 32*32 grid, to train the neural network in both cases. To evaluate, we run 16 test cases for both the two problems and the results are shown in Figure 4. We obtain an averaged MSE loss at 5.4e-14 for the first problem and 3.7e-3 for the second problem.

**Helmholtz equation** In this example, we test our model's performance by predicting the solution of Helmholtz equation $\nabla \cdot \nabla \boldsymbol{x} + \boldsymbol{x} = 0$. We set two different types of varying Dirichlet boundary conditions for this problem, one as $\boldsymbol{x} = -a/(p_u^2 + p_v^2)$ and another as $\boldsymbol{x} = a*sin((0.02p_u + 0.02p_v))$ with $a$ varies across the data. We use 4 data sample with respect to two types of boundary conditions with varying coefficients to train the neural network. For each type of boundary conditions, the training data is sampled on a $32 \times 32$ grid in two different domains respectively. The results are exhibited in Figure 5. In predicting solution,we achieve an averaged MSE of 5.25e-27 and 9.3e-29 in the two specific domains of the first type and For 3.5e-25 and 3.9e-18 for the second type.

**Wave equation** In this example, we demonstrate the ability of our Neural PDE solver by solving a time-dependent wave equation: $\nabla \cdot \nabla \boldsymbol{x} = \frac{\partial^2 \boldsymbol{x}}{\partial t^2}$ We use a standard finite difference method to generate the training data, which is 6-size data sample with each training sample indicating the target solution at the $n^{th}$ time step ($1 < n < 6$). Our model is trained to map $\boldsymbol{x}$ from the $n-1$ frame to the $n$ frame. The training data is sampled on a $49 \times 49$ grid with a source of $\boldsymbol{x} = sin(60(n*dt))$ at the center of the grid, where $dt$ is the time interval. With this observation of previous frames of a time-related wave function, our model is able to predict the following frames. The model's performance is tested by predicting the following 42 frames after the first 6 training frames. The training data and the predicting results are showed in Figure 13 and Figure 14. With an average MSE loss of 6.9e-4, we show that our model can precisely uncover the intrinsic structures of the kernel with sparse observation and can predict the numerical solution of it in a period of time.

**Navier-Stokes equation** We further demonstrate our model's ability in solving the Navier-Stokes equation:

$$\frac{\partial \vec{x}}{\partial t} + \vec{x} \cdot \nabla \vec{x} + \nabla p = \vec{f} \qquad \nabla \cdot \vec{x} = 0 \tag{9}$$

where, $\vec{x}$ stands for the velocity of the fluid, $p$ indicates the pressure and $f$ the body force. In each time step, our model is trained to accomplish the projection step which has a form of Poisson equation through the finite difference discretization method. The 6-size training data is sampled on a $32 \times 32$ grid with a fluid source in the left bottom corner of the domain, and the model is tested for 50 frames. The training data and the predicting results are showed in Figure12. With an averaged MSE loss of 4.09e-5, we show that our model can precisely uncover the intrinsic structures of the projection step in solving the Navier-stokes equation with sparse observation.

## 5.3 COMPARISON WITH OTHER METHODS

We compare our framework with a CNN baseline and PINN Lu et al. (2019).

**Comparison with CNN baseline** We evaluate our functional convolution model by comparing its performance with other naive convolutional neural network (CNN) structures in solving two typical problems targeted in this study: 1) 1D Poisson problem and 2) 2D time-dependent wave equation. To solve the 1D Poisson equation, we set up the CNN baseline structure as a 5-layer network consisting of three 1D convolution layers with a ReLU layer in between each two. The 2D time-dependent wave equation is specified by Equation $\nabla \cdot \nabla \boldsymbol{x} = \frac{\partial^2 \boldsymbol{x}}{\partial t^2}$. The CNN baseline structure is a 5-layer network, which is described in Table 1. Figure 7 shows the results. The figure shows that our framework converges fast and reduce the loss dramatically compared with the baseline. The details of the experiment could be found in Section B.1 in Appendix.

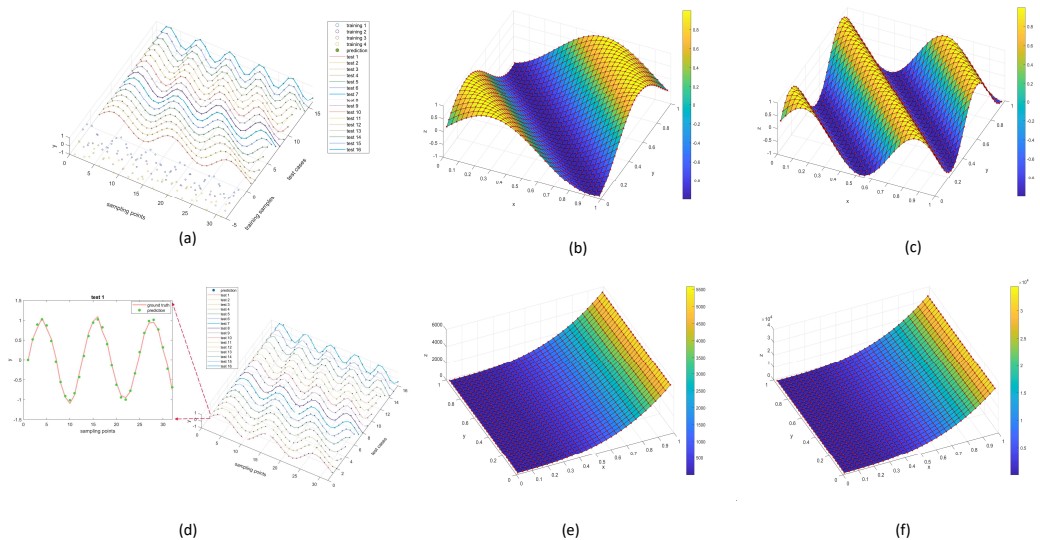

Figure 4: Test cases of 1D and 2D Poisson equations. (a)Test cases for predicting a sine function. (d)Test cases for predicting a sine function with $10\%$ noise. (b-f)Test cases in predicting the 2D Poisson equation with analytical solution (b,c) $\boldsymbol{x} = sin(a * p_u + a * p_v)$, (e,f) $\boldsymbol{x} = (ap_u)^3 + ap_v^2$.

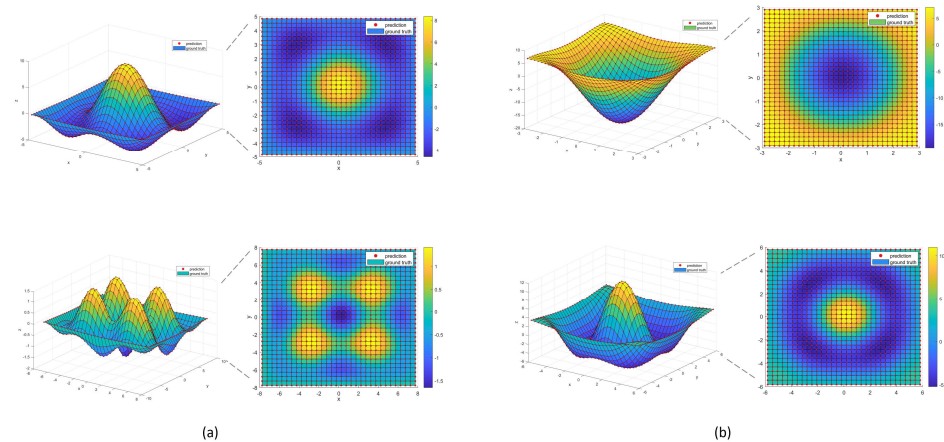

Figure 5: Test cases to predict the solution of a 2D Helmholtz equation with different boundary conditions in different domains. Right figures show the top view of the solution.

**Comparison with PINN**   We compare our framework with PINN Lu et al. (2019) in the setting of Helmholtz equation system. Specifically, the comparison is conducted with the Helmholtz equation $\nabla \cdot \nabla \boldsymbol{x} + \boldsymbol{x} = 0$. The comparison uses the Dirichlet boundary of $\boldsymbol{x} = -1/(p_u^2 + p_v^2)$. Both our framework and PINN are trained and tested on a $32 \times 32$ grid. Figure 8 shows the prediction results for PINN and our framework. Our framework achieves MSE of $6.05e - 15$, while PINN achieves MSE of $1.66e - 6$. The details of this experiment could be found in Section B.2 in Appendix.

## 6   RELATED WORKS

**PDE networks**   Long et al. (2018a; 2019) explore using neural networks to solve the Partial differential equations (PDEs). Li & Shi (2017) formulate the ResNet as a control problem of PDEs on manifold. Raissi et al. (2019) embeds physical priors in neural networks to solve the nonlinear PDEs. Han et al. (2018) handles general high-dimensional parabolic PDEs. Brunton et al. (2020) gives an overview of neural networks in turbulence applications. Wu et al. (2020) train generative adversarial

networks to model chaotic dynamical systems. Han et al. (2016) solves high-dimensional stochastic control problems based on Monte-Carlo sampling. Raissi (2018) approximate a deterministic function of time and space by a deep neural network for backward stochastic differential equations with random terminal condition to be satisfied. Wang et al. (2019) first propose to use reinforcement learning (RL) to aid the process of solving the conservation laws. Jagtap et al. (2020b) propose a neural network on discrete domains for nonlinear conservation laws. Jagtap et al. (2020a) employ adaptive activation functions to solve PDEs and approximate smooth and discontinuous functions. Pang et al. (2019) combines neural networks and Gaussian process to solve PDEs.

**Prior-embedded neural simulators**   Many recent learning physics works are based on building networks to describe interactions among objects or components (see Battaglia et al. (2018) for a survey). The pioneering works done by Battaglia et al. (2016) and Chang et al. (2017) predict different particle dynamics such as N-body by learning the pairwise interactions. Following this, the interaction networks are enhanced by a graph network by Kipf et al. (2017) for different applications. Specialized hierarchical representations by Mrowca et al. (2018), residual corrections by Ajay et al. (2018), propagation mechanisms by Li et al. (2019), linear transitions by Li et al. (2020) were employed to reason various physical systems. On another front, modeling neural networks under a dynamic system's perspective has drawn increasingly attention. In 2018 Chen et al. (2018) solves the neural networks as ordinary differential equations (ODEs).

## 7   CONCLUSION

In this paper, we introduced neural PDE, a machine learning approach to learn the intrinsic properties of PDEs by learning and employing a novel functional convolution and adjoint equations to enable the end-to-end training. Our model resents strong robustness against the arbitrary noise. The main limitation of our work is that we assume the PDE systems are sparse. That is, the relation is restricted locally. To enable the constrain in larger area, one can enlarge the kernel size, but this can potentially cost much more computation and memory. For the future applications, we plan to apply the method to solve real-world 3D applications, such as incompressible fluids and nonlinear soft materials. We also plan to scale up the system by leveraging the sparse iterative linear solver for solving the linearized PDEs in each iteration.

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

## A    ABLATION TESTS

The neural networks are trained by a hybrid method of IpOpt and Adam optimizer. Here we demonstrate our hybrid optimizer's capability to outperform a typical Adam optimizer and a typical SGDM optimizer, which is shown in Figure 6.

## B    COMPARISON WITH OTHER METHODS

### B.1    COMPARISON WITH NAIVE CNN STRUCTURE

We evaluate our functional convolution model by comparing its performance with other naive convolutional neural network structures in solving two typical problems targeted in this study. For the first case to solve the 1D Poisson equation, we set up the baseline structure as a 5-layer network

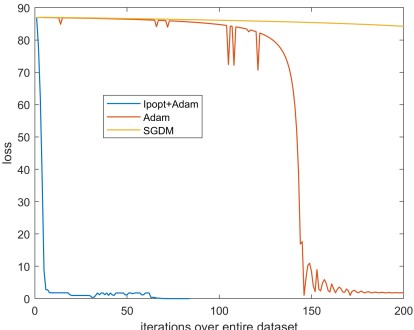 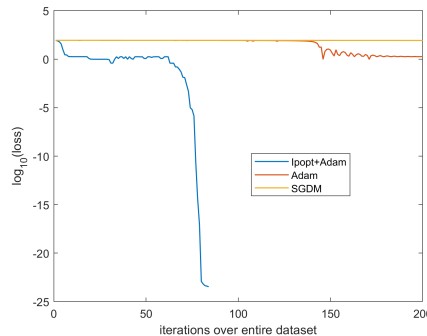

Figure 6: Training our model to predict the solution for the 1D Poisson problem $\nabla \cdot \nabla x = 0$ using different optimization methods. The IpOpt+Adam optimizer stops at early converge with the total loss of 8.5e-25.

consisting of three 1D convolution layers with a ReLU layer in between each two. The second problem is the 2D time-dependent wave equation specified by Equation $\nabla \cdot \nabla x = \frac{\partial^2 x}{\partial t^2}$. The baseline for this problem is also set to be a 5-layer network, which is described in Table 1. For each of the baselines, the input is set to be the right hand side value and the output is the predicted solution. The results are shown in Figure 7, which demonstrates that our functional convolution model outperforms the naive convolutional neural network both in accuracy and converging rate.

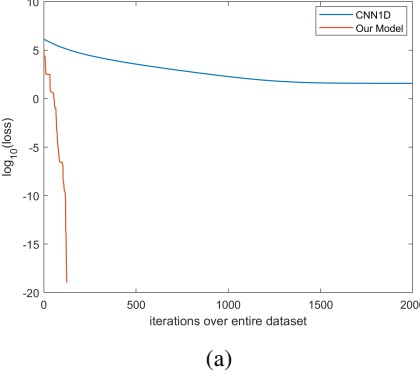 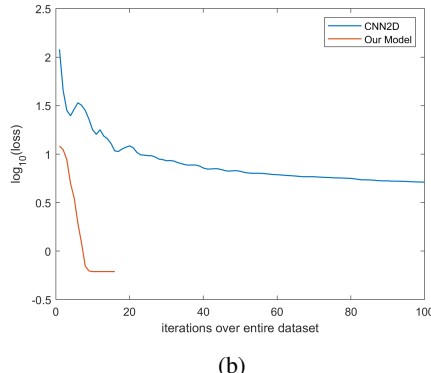

(a)             (b)

Figure 7: comparison between predictions from Naive CNN networks and our model to predict the solution for (a) the 1D Poisson problem $\nabla \cdot \nabla x = 0$ and (b) the time-dependent wave equation.

### B.2 COMPARISON WITH PINN

We compare our framework with PINN Lu et al. (2019) in the setting of Helmholtz equation system. Both models are trained and tested in $32 \times 32$ grid. Figure 8 shows the prediction results for PINN and our framework.

## C 1D EXAMPLES

### C.1 1D POISSON EXAMPLE

We employ our functional convolutional model with the aim to solve typical 1D Poisson problems with constant kernels, the testing results of which are shown in Figure 9.

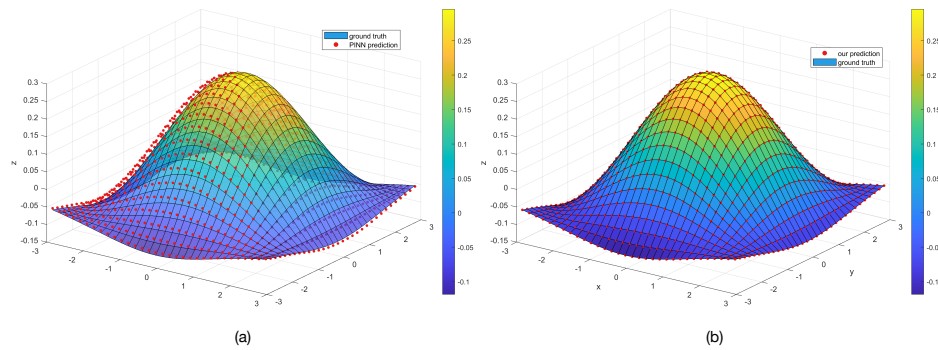

Figure 8: Comparison with PINN. Figure (a) shows the predicted value of PINN with MSE of 1.66e-6. Figure (b) shows the predicted value of our framework with MSE of 6.05e-15.

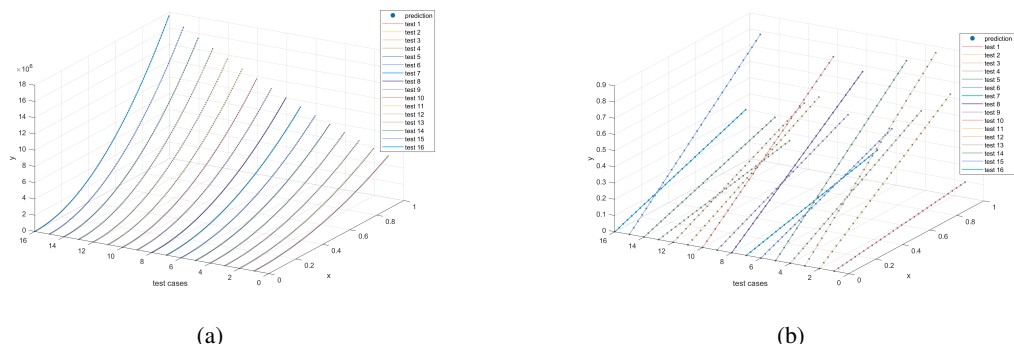

Figure 9: Two test cases for predicting the solution of a 1D Poisson equation $\nabla \cdot \nabla \boldsymbol{x} = b$ with (a) target solution $x = (ap)^2$ and (b) $b = 0$. The dots represent the predicted values and the curves indicate the true values.

## C.2  1D STABILITY TEST WITH NOISY INPUT

We conduct a stability test by adding noise values to the setting of the 1D constant kernel experiments. We test our framework with various scales of noise from $[-.1, .1]$ to $[-.35, .35]$ with step size of .05 with respect to the extreme values in the target solution. We compare our framework with denoised solutions in Figure 10. The figure shows the framework can automatically obtain precise solution even though our framework cannot access accurate solutions during the training procedure.

## C.3  1D SPATIAL VARYING AND NONLINEAR EXAMPLE

We also train the model to predict the solution of $\nabla \cdot (1 + |\pi p|)\nabla \boldsymbol{x} = 0$ and $\nabla \cdot (1 + |x| + sin(|x| * .001))\nabla x = 0$. The results are shown in Figure 11.

## C.4  2D NAVIER STOKES EXAMPLE

The 6-size training data is sampled on a $32 \times 32$ grid with a fluid source in the left bottom corner of the domain, and the model is tested for 50 frames. The functional convolution network is set to be $3 \times 6 \times 6 \times 6 \times 6 \times 3$. We show the ground truth by a typical fluid solver, our prediction, and the absolute error between the two of Frame 1, 25, and 50 in Figure 12.

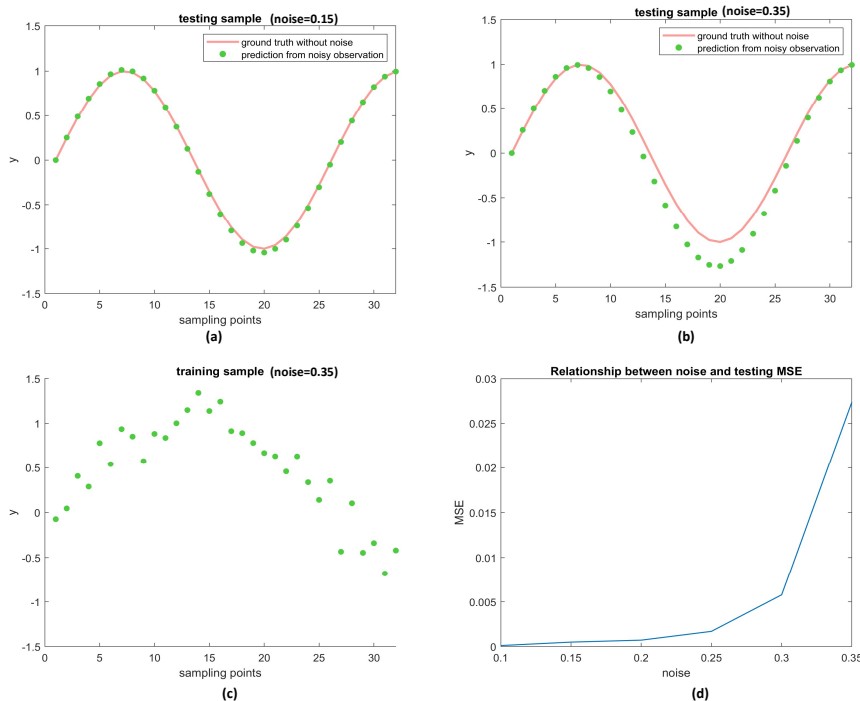

Figure 10: Figure (a) and (b) show two testing cases with noise of .15 and .35 with respect to the extreme values in the same target solutions. The MSE errors of the 16 testing cases for them are 0.0005 and 0.0274. Figure (c) shows one training sample with noise. Figure (d) shows the relationship between the noise in the observation and testing MSE.

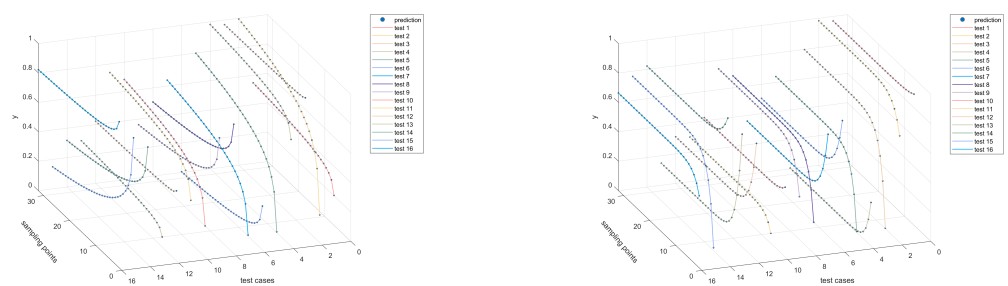

Figure 11: Test cases for predicting the numerical solution of an unknown Poisson PDE. The left figure shows function domain of all tests to predict the Poisson PDE with with spatially varying coefficients. The right figure shows tests to predict the solution of a nonlinear PDE specified by $\nabla \cdot (1 + |x| + sin(|x| * .001))\nabla x = 0$.

## D   WAVE EQUATION

Figure 13 shows the training data of the wave equation. Figure 14 shows the predicted solution of time dependent wave equation.

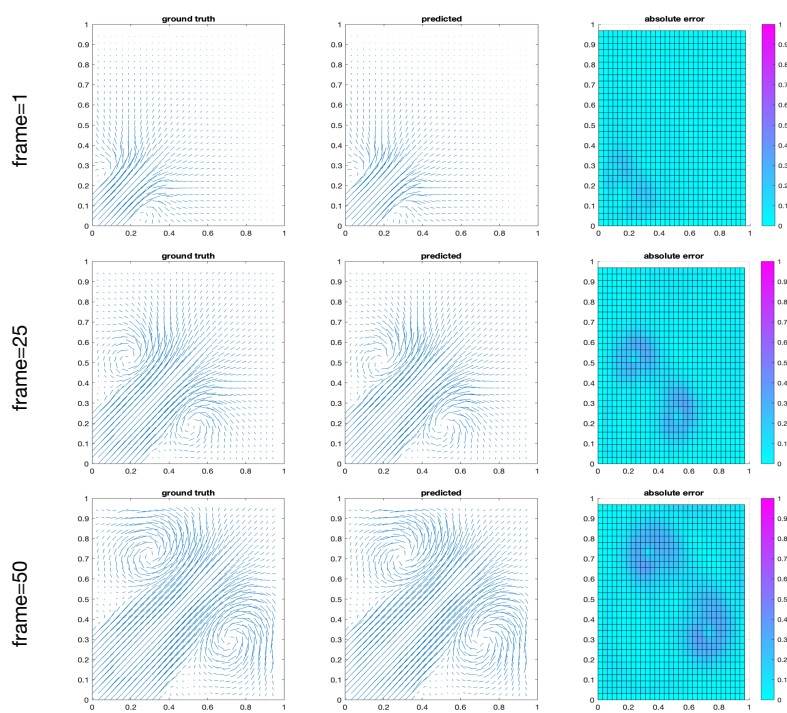

Figure 12: 2D Navier Stokes example shows the ground truth by a typical fluid solver, our prediction, and the absolute error.

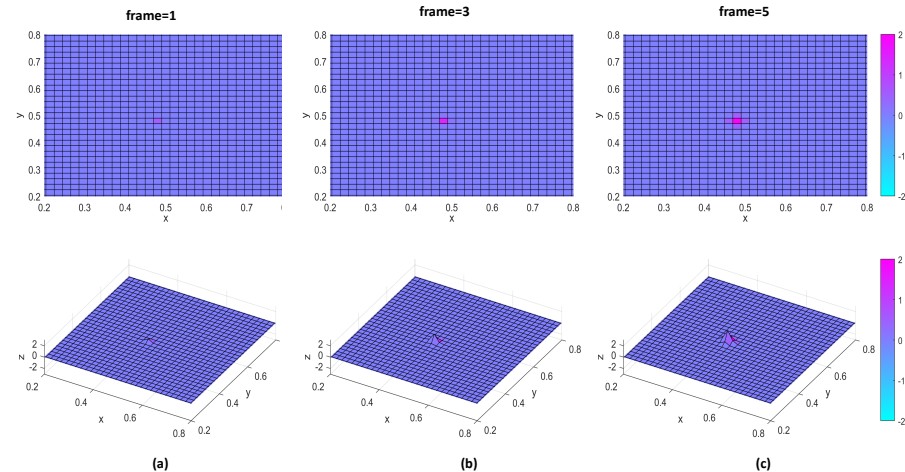

Figure 13: The training data to solve the time-dependent wave equation specified by Equation $\nabla \cdot \nabla x = \frac{\partial^2 x}{\partial t^2}$. (a-c) The target solution at timestep 1, 3, 5, with top view and 3D view. The equation is solved in $domain = [0, 1] \times [0, 1]$, here we only show the plots in $[0.2, 0.8] \times [0.2, 0.8]$.

# E    DETAILS ON NEURAL NETWORK STRUCTURES

We show the details of the naive CNN structures in Table 1 which are trained as baseline compared to our convolution model. We also refer readers to Table 2 for the details of neural network structure and training parameters across different examples.

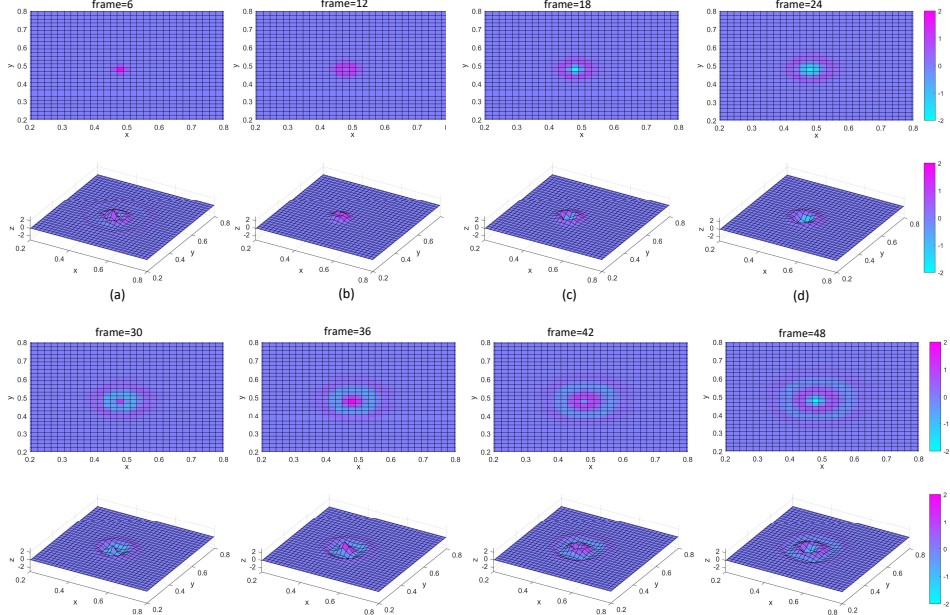

Figure 14: The predicted solution of the time-dependent wave equation. (a-h) The predicted solution at timestep 6, 12, 18, 24, 30, 36, 42, 48 with top view and 3D view. The equation is solved in $domain = [0, 1] \times [0, 1]$, here we only show the plots in $[0.2, 0.8] \times [0.2, 0.8]$.

Table 1: Naive CNN baselines

| Baseline | neural network structure | grid | # training smaples |
|---|---|---|---|
| 1D CNN structure | cnn1d(in=1,out=1,kernel=3), ReLU, cnn(in=1,out=1,kernel=3),ReLU, cnn1d(in=1,out=1,kernel=3) | $16 \times 1$ | 4 |
| 2D CNN structure | cnn1d(in=1,out=2,kernel=3), ReLU, cnn(in=2,out=2,kernel=3),ReLU, cnn1d(in=2,out=1,kernel=3) | $20 \times 20$ | 8 |

Table 2: Neural network structure and training parameters

| Equation | Dimension | Network | # Parameters | # Training sample | # Test sample | Test MSE |
|---|---|---|---|---|---|---|
| $\nabla \cdot \nabla \boldsymbol{x} = b$ with target solution $x = (ap)^2$ | $128 \times 1$ | $3 \times 4 \times 4 \times 3$ | 31 | 4 | 16 | 8.3e-20 |
| $\nabla \cdot \nabla \boldsymbol{x} = 0$ | $32 \times 1$ | $3 \times 4 \times 4 \times 3$ | 31 | 4 | 16 | 5.0e-29 |
| $\nabla \cdot \nabla \boldsymbol{x} = b$ with target solution $x = sin(ap)$ | $32 \times 1$ | $3 \times 4 \times 4 \times 3$ | 31 | 2 | 16 | 9.8e-4 |
| $\nabla \cdot \nabla \boldsymbol{x} = b$ with target solution $x = sin(ap)$ with noise | $32 \times 1$ | $3 \times 4 \times 4 \times 3$ | 31 | 2 | 16 | from 5e-4 to 2.7e-2 |
| $\nabla \cdot (1 + |\pi p|)\nabla \boldsymbol{x} = 0$ | $32 \times 1$ | $6 \times 6 \times 6 \times 6 \times 3$ | 108 | 4 | 16 | 1.3e-6 |
| $\nabla \cdot (1 + |x| + sin(|x| * .001))\nabla x = 0$ | $32 \times 1$ | $3 \times 6 \times 6 \times 3$ | 45 | 4 | 16 | 2.1e-5 |
| $\nabla \cdot \nabla \boldsymbol{x} = b$ with target solution $x = (ap_u)^3 + ap_v^2$ | $32 \times 32$ | $3 \times 14 \times 14 \times 14 \times 14 \times 3$ | 325 | 4 | 16 | 5.4e-14 |
| $\nabla \cdot \nabla \boldsymbol{x} = b$ with target solution $x = sin(ap_u + ap_v)$ | $32 \times 32$ | $3 \times 10 \times 10 \times 10 \times 10 \times 10 \times 3$ | 293 | 4 | 16 | 3.7e-3 |
| $\nabla \cdot \nabla \boldsymbol{x} + \boldsymbol{x} = 0$ with boundary $\boldsymbol{x} = a/(p_u^2 + p_v^2)$ | $32 \times 32$ | $3 \times 5 \times 5 \times 5 \times 5 \times 3$ | 68 | 4 | 16 | 5.3e-27 |
| $\nabla \cdot \nabla \boldsymbol{x} + \boldsymbol{x} = 0$ with boundary, $\boldsymbol{x} = a * sin((0.02p_u + 0.02p_v))$ | $32 \times 32$ | $3 \times 5 \times 5 \times 5 \times 5 \times 3$ | 68 | 4 | 16 | 9.3e-29 |
| $\nabla \cdot \nabla \boldsymbol{x} = \frac{\partial^2 \boldsymbol{x}}{\partial t^2}$ with center source $\boldsymbol{x} = sin(60(n * dt))$ | $49 \times 49$ | $3 \times 5 \times 5 \times 5 \times 5 \times 3$ | 68 | 6 | 42 | 6.9e-4 |
| $\frac{\partial \vec{x}}{\partial t} + \vec{x}\nabla \vec{x} + \mu \Delta \vec{x} + \nabla p = \vec{f}$ | $32 \times 32$ | $3 \times 6 \times 6 \times 6 \times 6 \times 3$ | 87 | 6 | 50 | 4.09e-5 |

