# OpenReview forum: "Neural Partial Differential Equations with Functional Convolution"
_ICLR.cc/2021/Conference — Reject_

### Official Review · AnonReviewer3 · 2020-10-22

**Rating:** 4
**Confidence:** 4

**Review:**

Post-discussion update: The authors only partially adressed my concerns in their rebuttal. The paper suffers from lack of comparisons: only 2 baselines are compared, and only on few systems. Crucially the new Navier-Stokes experiment lacks comparisons. The authors also couldn't respond to my questions about research context or scope: it's difficult to assess what this work actually claims in relation to competing methods. For a machine learning paper this is not enough.

-----

The paper proposes input-dependent convolutions for PDE learning under Picard solvers. The main contribution seems to be the spatially non-stationary filters, which is a useful and somewhat known CNN technique. It is easy to see how this makes the neural PDEs more powerful, but it’s unclear in which setting the spatially-evolving differentials are warranted in PDEs. The paper should discuss the motivation and justification of this choice more.

It’s unclear if the adjoints or picard solver are novel or adaptations from earlier works.

I wonder why the spatial convolutions do not seem to overfit. They can learn in principle arbitrarily complex mappings, and with scarce data should overfit badly. How was this handled?

The paper is written in a clear manner, but almost all math suffers from undefined symbols and variables, which makes the math frustrating to read. Also the problem domain and problem definitions are undefined.

The experiments show that the method perfectly learns example systems with practically 0 test error. While this sounds great, this also raises a lot of questions. Perhaps the problems were trivial to begin with (there are no comparison to other methods), or maybe they do not generalize (there are no extrapolation experiments to regions where no data has been observed). The problems were also very small (1D or 2D), so are they practically relevant or do they actually require neural machinery of this kind? The author’s should present learning curves and ablation studies to show when how far the method can extend before breaking (fewer data, more noise, more dimensions, more complex systems, etc), and include uncertainty analyses. Extensive comparisons to other methods (both neural and non-neural) are necessary to demonstrate useful contribution. Finally, it’s unclear if these results indicate generalisation (in some sense).

All experiments seem almost textbook examples of simple, regular and clean cases. Neural networks excel when applied to messy and complex problems. Why are these a good application of neural PDEs?

The paper presents an incremental convolution extension to learning of PDE convolutions in Picard setting. The results show fantastic performance, which however is undermined by lack of any comparisons and the almost toy-likely simple systems that were studied.



Technical comments
o The functional C is undefined and unexplained, what does it mean? It seems to be a “form” instead of differential. How does C relate to PDEs?
o Also “A” and “b" are explained, how do they relate to PDEs?
o The notation throughout needs to be better defined, eg. domains of all variables
o The paper is lacking the basic PDE definitions completely. Please define the problem this paper is tackling
o Clarify what is “PDE unknown x”. I assume it's the solution.
o What is “current state of x”? Isn’t “x” the state itself?
o eq1 is not a function of “p” but it still uses “p”, please fix
o eq1 seems to redefine C(x,p) to C({x},{p}). How do we handle neighborhoods? How are the neighborhoods defined (sets/subgraphs/tensors?) What does the [..](pj) notation mean? How come we have x^t here, despite A(..) not being a function of time. What is the domain/codomain of C? Please clarify
o Please define x_0,x (is it a matrix?)
o what is “feature map” in eq 2?
o define “w” in eq2
o I wonder if w(x_mn) would have been more intuitive notation for input-dependent w-functions
o N = {} is either a set or flattened vector, not both
o It would be more sensible to define N and C as matrices than vectors since they are tensors with two indices
o what is x_n (is “n” a time or something else?)
o Is the “A” in eq5 the “A” in eq 2 or 3? Where is the “b” coming from, it’s not defined
o Is the adjoint derivation 2.3. novel?
o What is the “MSE” in experiments? Please report training and testing errors separately
o I don’t understand why a time-dependent wave equation was tested, since the model is not time-dependent but spatially dependent.

---

### Official Review · AnonReviewer4 · 2020-10-27
**Seems similar to previous works. More experiments maybe needed.**

**Rating:** 5
**Confidence:** 4

**Review:**

Summary: the work proposes to use neural networks to learn a kernel C(x,p) for PDEs. It embeds the neural network into iterative solvers and trains it with the Adjoint method.

The writing is clear in general but notation-heavy. For example, it could be better to define notations such as $A$ before using it.

Strong points:
- Clear,
- Well-motivated,
- Theoretically solid,
- The example given on page 3 is quite helpful.

Concerns:
- The idea seems similar to the previous works.
- Lack of comparison and benchmarking.

I am not very certain about evaluating the novelty of this work. The idea to approximate the kernel with the neural networks has been proposed and studied in (https://arxiv.org/abs/2003.03485, https://arxiv.org/abs/2010.08895). The major difference seems to be that they directly learn solution operators, while in this work we embed it into iterative solvers.
It will be great if the authors can help me understand the contribution beyond the previous works.

Another concern is about the experiments. The test equations (Poisson, Helmholtz, Wave Equation) presented in the paper seems fairly simple. I wonder how other methods such as PINN, or the numerical solver perform on these equations. It will be great to have some benchmarks and comparisons with existing works. It can help me better evaluate the performance of the method.

Also, I found the scale for the neural network is very small. The authors claim larger networks are not needed, but it's better to have some justification or experiments.

Questions:
1. how does it differ from the existing work?
2. how does it empirically compare with numerical solvers or other deep learning-based methods?
3. is it possible to try large networks in the experiments?

Recommendation:
In general, I found this work interesting and concrete, but I am not certain about the novelty. Therefore, I would like to put this paper on margin.

---
_Updated review:_

>The updated manuscript has some substantial improvements.
>
> I feel the biggest problem is that the authors didn't clearly state the problem settings. If I understand correctly, in their framework the equation is fixed but unknown. The training data are several points in the domain (with parameters input) and testing data are other points. So basically, we doing interpolations. But even the PDE is unknown, they do assume some structure of the PDE, I think.
>
> Other PDEs frameworks are either 1. solver-type: the equation is known and fixed, they directly solve for the solutions. 2. operator-type, the equations are unknown and changing. Train on inputs-outputs for several equations, and test on others. Their setting is quite different. I guess it's the reason their performance is much better in the updated comparison. On the other hand, it's also hard to evaluate their performance since there are no fair benchmarks.

> In general, I feel this paper is novel and concrete, while it's not very complete and well-presented. I agree with other reviewers that this paper is not ready to publish.

---

### Official Review · AnonReviewer1 · 2020-10-28
**Paper needs rewriting to be understandable & reproducable**

**Rating:** 4
**Confidence:** 3

**Review:**

Summary:
The paper proposes a neural network based solver for PDEs based on the Picard Iteration. In the numerical experiments section the paper applies the method to solve 1d or 2d PDEs.

Disclaimer:
I am not an expert in PDE solvers but I am researching topics related to your frequently cited works of Kipf et. al., Battaglia et. al. & Raissi et. al.. Therefore, my knowledge on PDE solver might be a limiting factor to understand the complete work.

Review:
For me personally it is really hard to parse the paper and get a rough understanding of the paper. I cannot fully understand the application, why it should be better to numerical solvers, how it roughly works, what kind of training data is used. By now I have reconstructed the genreal idea from the details presented but I am uncertain about whether my understanding is correct and it is so limited that reproduction of the algorithm is not possible. Furthermore, the reader should not be required to reconstruct the idea from a paper from the details.
Therefore, an evaluation of the contribution and significance is not really possible for me. As I cannot **even** get a rough understanding from reading the paper and googling related work / textbooks, I think the paper needs to be rewritten to make it suitable for the ICLR community. If I understood the approach partially correctly, I think that the idea is really neat, interesting and novel and gave me a new perspective. However, the experiments also focus on well posed standard problems and not on the more complex problems of multi-body contact right mechanics which is targeted with many of the referred works.

In the following, I will state my main questions regarding the understanding.


Summary & Questions for each section:

(1) Introduction:
For me the introduction is missing to state the clear problem statement and the available information to solve the problem. What kind of application do you have in mind? The paper cites a lot of graph networks from Battaglia et. al. or Kipf et. al. but I cannot relate your problem to the works of these authors. I guess you also want to learn a simulator but then the experiments just focus on solving very small PDEs. Besides the main motivation I am missing an exact mathematical problem statement. My guess is that you want to solve for x(p) given the differential equation described by \partial x / \partial p = f(p, x(p)). Now the question for me is, what is known / unknown about the differential equation? E.g.,
* Do I assume to know the structure of f but not the exact parameters?
* Do I know f but not the solution x(p)?
* Do I know x(p) at some points and nothing about f?

My guess is the last, but I am uncertain. Following a precise problem statement, I would like to see the general approach how to solve the problem and how this is currently done. For this paper my guess is that the standard approach to obtain the PDE solutions is to derive the filter C manually from the analytical f. Then one can solve for the solution of x(p), by convolving C with the mesh of x_k obtaining A and obtaining new x_{k+1} by solving A x_{k+1} = b. Is that correct? Furthermore, I am uncertain about the definitions of b and x. I guess x is a vectorized mesh of the domain of x. Such an short theoretical overview of standard approaches would be very helpful as the google results of picard iteration lead to very different symbolic approaches.

(2) Method:
I would like to understand the general algorithm first and then deep dive into the details. The current version just presents the details but not the overall approach. Hence the reader has to reconstruct the main approach from the details, which is really cumbersome. My current understanding of your approach is that you start with a random neural network for C and perform the picard iteration with the random C. At the end of the picard iteration you compute the MSE of the predicted solution and the observed solution and update the filter C to minimize the mse at the end of the picard iteration. To compute the derivatives about the iterative picard solver you use the adjoint method. I am not certain if I understood it correctly or not.

(3) Network Architecture:
The paper uses a uncommon optimization scheme for deep networks, i.e., IPopt + Adam. And the experiments show that pure Adam does not work, could the authors please discuss the differences between IpOPT and Adam and discuss why pure Adam does not work?

(4) Experiments:
The section provides an overview about simple 1d and 2d PDE problems. The data samples are really small 4 - 6 samples (if I understand correctly) and the obtained performance is outrageously good mse, i.e., 10^-14. This seems a bit fishy to me and proposes that the experiments are way too simple and the provided data is way too perfect. The problem with real-world data is that it is really noisy and one would need to recover the structure despite the noise. Therefore, I would like to see this approach applied to much more challenging problems, preferably with real data.

(5) Related Work:
As mentioned before the connections to other works are not really made clear, especially to the application centric graph networks.

One more general question:
If I understand the motivation correctly, is this approach also applicable to controlled systems as the controller would change the dynamics frequently and then the filter C would need to be relearned in each iteration of the policy optimization?


**Post Rebuttal Comments**: The authors improved the paper during the rebuttal but the clarity is not sufficient and the results are still puzzling. I do not fully understand how one can learn the perfect solution with only 3-4 data points.

---

### Official Review · AnonReviewer2 · 2020-11-02
**Recommend accept**

**Rating:** 4
**Confidence:** 2

**Review:**

ICLR Neural PDEs

Summary:

This paper aims to use neural networks to find hidden structure in PDEs and predict their solution. Their neural network examples are extremely small (up to 325 parameters) and require little data (up to 8 samples). Instead of a classical convolution, which combines the neighbors with fixed weights, they learn a "functional convolution" that combines the neighbors with weights that are themselves functions of the neighbors. This takes advantage of "translational symmetry" in discretized differential operators. The tiny network is then embedded in a Picard forward iterative procedure. An adjoint backward gradient calculator relies on being able to do auto-differentiation in neural networks. They specifically consider elliptic boundary value problems. It is key that the PDE systems are sparse (dependent on a limited number of neighbors).

Strong points:

I am very impressed by the small networks and limited training data.
Some of the test MSEs are quite impressive.
To my (albeit imperfect) knowledge of the literature on machine learning for PDEs, this is quite creative. It also seems like a big step forward as far as low error with limited data.
It's nice to see the variety in the six test cases.

Weak points/Clarification questions:

This paper needs more details on the training and testing data.
- What counts as one data sample?
- The input & output is not always clear.
- How much variance is there within the training & test data? For example, for the "constant kernels" case, I can tell from Figure 8. However, for the "spatially varying coefficients" case, I think we never see the data. Also, for some cases, I don't think it's mentioned what is varied between examples.
- It's sometimes unclear from the captions whether we're seeing training data or test data, or whether it's the true data or the results.
- I'd like to see both training & test error to see if there is overfitting.
- I don't see any mention of validation data, which is often what people use to choose hyperparameters. What did you use to choose hyperparameters? If the test data was used for this, we need new test examples, as test examples need to be held out until it's time to report errors.

I may have missed this, but about how long does it take a train your method, and how long does it take to use the trained network for prediction?

The text in the figures are often too small to see. All parts of Figure 3 (a) & (d) are difficult to see.
The Related Work section (Section 5) lists a lot of papers, but doesn't explain what advantages your paper has.

~~~~~
Update:

I think that the revised paper is an improvement,  but it's not ready. I think there is still missing information to make it clear what you did (as the other reviewers have commented as well.) I am particularly concerned that we still don't have a comparison of train & test errors for each network, and that we don't still know which dataset was used for selecting hyperparameters (train vs. val vs. test). These are crucial questions for deep learning results, and I always check this when I peer-review. I reduced my rating based on these concerns.

---

### Author Response · Authors · 2020-11-25
**Rebuttal and Revised Submission (part 1)**

Dear Reviewers,

Thank you for  the comments and the valuable feedback. We are very pleased to address your concerns and incorporate your suggestions by making several main updates (see the bullets below) in the revised manuscript. All the changes are colored as blue in the new version.

### Main changes
- We added a motivating example section to better illustrate the background of numerical PDE (including the linear Poisson and nonlinear Picard). (Section 2, Figure 1)
- We made all the suggested expositional changes regarding the equations, symbols, and descriptions. (Section 1 with two inset figures and Section 2 naming convention)
- More comparison with state-of-art PDE solvers (PINN in Section 5.3, B2 and Figure 8)
- Stability test with noise (Section 5.1, Section C2, Figure 10)
- An additional example showing complex systems (Navier-Stokes equation, see Section 5.2, C4 and Figure 12)
- High-resolution figures (we updated the quality of all the figures)

### Compared with the SOTA (R1,R2,R3,R4)
We want to highlight that our method targets at solving an inverse problem by uncovering an effective and compact representation (the structure of C) of an unknown PDE system from scarce observation data. This deviates from the vast majority of the existing neural PDE solvers or the numerical PDE solvers, which all assume a known PDE description in hand. Therefore, our approach does not aim to improve an existing numerical paradigm to find more accurate or efficient numerical solutions (for example, PINN, PDE-Net or other PDE solvers). Instead, we strive to solve a PDE from sparse observation without knowing its expression. Mathematically, our model aims to reconstruct the entire solution manifold of a series of PDE problems that have the same hidden intrinsic structure (the same pattern of C).

### Picard solver (R2,R3,R4)
Incorporating mathematical or numerical inductive priors into a network architecture has been emerging as an effective way to tackle the learning problems’ nonlinearity. A famous example in the community is the Koopman operator [https://doi.org/10.1073/pnas.1517384113], which embeds the prior of iterative linearization of a nonlinear dynamic system by constructing a data-driven paradigm that can linearize its evolution. Our approach of employing the Picard interaction follows the same philosophy. We introduced a combo of classical numerical tools (adjoint solver+Picard iteration) and modified them to accommodate data-driven system identification, which turns out to be extremely effective in uncovering nonlinear PDE structures. The Picard iteration method is a well-established scheme in numerical mathematics, and as far as we know, our method is the first to introduce it to facilitate data-driven applications.

### Simple PDE forms (R2,R3,R4)
We want to emphasize that the complexity of a PDE system (in particular, its solution manifold) is hidden behind its mathematical form. In many cases, the simple form of an equation does NOT indicate a trivial solution. A typical example is the difference between the Poisson and the Helmholtz equations: they share very similar expressions (with only a different +/- symbol) yet their solutions’ distribution, frequency, and numerical difficulties on convergence are completely different. We believe the numerical tests we conducted well covered the broad spectrum of the common PDEs, in both 1D and 2D, with a particular focus on elliptical, nonlinear systems (which is a rarely explored domain in the scientific ML community). On another hand, we want to emphasize that the mathematical form of the target PDEs is completely hidden to our learning algorithm. Our algorithm did not leverage any prior information from the equation itself to reason its solution structure, which separates the network and the PDE clearly. Last, to further demonstrate the complexity of our target systems, we added an experiment for solving the dynamic Navier-Stokes equations (see Section 5.2) in our updated manuscript.

### Robustness with messy data (R2,R4)
We conducted tests with noise in Section 5.1 in the previous manuscript. To strengthen this part, we added another experiment to test our model’s robustness in more drastically noisy settings (see Section 5.1). The test shows that our framework is able to denoise without even knowing the clean solution throughout the training. The reason is that by fitting the parameters in C to minimize the loss, despite that the loss is affected by noise, our model is still able to uncover the hidden structure of matrix A’s kernel.

---

### Author Response · Authors · 2020-11-25
**Rebuttal and Revised Submission (part 2)**

Other minor comments:

### Reviewer #1:

**What counts as one data sample? How much variance is there within the training & test data?**

A data sample is a mesh grid. In our experiments, a data sample can be a $32\times1$, $128\times1$ grid (1D examples) or $32\times32$, $49\times49$ grid (2D examples). Different training data samples are sampled on different target solutions. For example, in the spatially varying case, the 4-size training data is sampled on solutions of 4 randomly generated boundary conditions (for each data sample, $x_0=random(0,1)$ and $x_{31}=random(0,1)$, where $x_0$ and $x_{31}$ are the two boundary points), and the 16 testing case is tested on the target solution with also random generated boundary conditions.

**I would like to see both training data and testing data to see if there is overfitting.**

For your reference, the MSE of training data from IpOpt + Adam optimizer in Figure 6 is 8.5e-25, and the MSE of 16 testing data is 5.1e-25.

**How to choose hyperparameters for NN?**

We use a grid search method within a scale of layers and number of neurons in each layer.

**How long does it take a train in your method, and how long does it take to use the trained network for prediction?**

For 1D cases, a train can be finished within several seconds. For 2D cases, a train takes less than half an hour to finish for the examples with $32\times32$ grid and 2 hours for the examples with $49\times49$ grid. A prediction always takes several seconds to finish.

### Reviewer #2:

**Questions regarding understanding**

You do understand our method in the correct way. As you suggested, we clearly clarify the target problem Section 1 and add motivation in Section 2, together with two inset illustration figures and Figure 1 to better state the problem and our method.

**IpOpt and Adam?**

We use IpOpt to accelerate the exploration of the accurate local minimum and Adam to enhance its global search ability. IpOpt is good at fast converging but is vulnerable to bad initiation and easily falls into local minimum. Adam is a stable optimizer but takes relatively long time to converge. Furthermore, Adam dangles around a suboptimum in some of our cases, as mentioned in this paper [https://arxiv.org/abs/1904.09237v1]


**Is our model applicable to controlled systems?**

For PDE systems with a temporally constant C (such as a soft body system with elastic material model), our method is applicable. We cannot handle temporally varying C with the current scheme.


### Reviewer #3:

**Is it possible to try larger neural networks?**

Sure.With larger neural networks, the loss can be further reduced. Take the 2D Poisson case as an example, while the $3\times10\times10\times10\times10\times10\times10\times3$ NN obtains a testing MSE of 3.7e-3, the $3\times12\times12\times12\times12\times12\times12\times3$ NN can obtain a testing MSE of 3.2e-6. In most cases, however, the small NN has already reached high accuracy and larger NN can only lead to marginally reduced loss.

### Reviewer #4:

**Unclear definition of symbols**

As you suggested, we have modified the manuscript by providing clear definition of target problem, symbols and equations, such as section 1, 2 and Figure 1.

**Ablation tests**

We compare our method with results from other optimization methods which serves as an ablation test. For your concern, we also add new experiments to demonstrate our model’s performance with more noise (Section 5.1) and in complex system (Navier stokes equation in 5.2)

---

### Decision · Program_Chairs · 2021-01-07
**Final Decision**

**Decision:**

Reject

**Comment:**

The objective of the paper is to develop a framework for solving PDES with reduced model size and for scarce observation settings. It proposes to use functional input dependent convolutions for learning spatio-temporal differential operators together with a non linear numerical scheme (Picard solver). Training makes use of an adjoint formulation.

All the reviewers agree that the authors improved the initial version but opt for a reject. In its present form, the technical description is still incomplete with missing explanations. The experiments should be reinforced and the results are partly unexplained.